# Comparative Multi-Criteria Assessment of Hydrological Vulnerability—Case Study: Drainage Basins in the Northeast Region of Romania

**Andra-Cosmina Albulescu [1], Ionuț Minea [1],\*, Daniel Boicu [2] and Daniela Larion [1]**

[1]  Faculty of Geography and Geology, Alexandru Ioan Cuza University of Iasi, 700505 Iasi, Romania; cosminaalbulescu@yahoo.com (A.-C.A.); danielalarion@yahoo.co.uk (D.L.)

[2]  Research Center with Integrated Techniques for Atmospheric Aerosol Investigation in Romania, RECENT AIR, Laboratory of Interdisciplinary Research in Geo-Chemistry of Rural Areas, Environmental Quality Monitoring Station for Geographic Research, Alexandru Ioan Cuza University of Iasi, 700505 Iasi, Romania; boicu.d.daniel@gmail.com

\*  Correspondence: ionutminea1979@yahoo.com

**Abstract:** Hydrological vulnerability (HV) is a (changing) underlying condition in all drainage basins, depending on the dynamics of the potentially dangerous hydrological phenomena, the particularities of drainage networks, land use patterns, and processes that shape landforms in extensive periods. The socioeconomic attributes and the hydrotechnical infrastructure add up to the manifestation of this type of vulnerability. In this paper, we assess the HV levels of 81 drainage basins in the NE of Romania for three distinctive periods (1990–1999, 2000–2009, 2010–2018), using a multi-criteria approach. Two classical multi-criteria decision making (MCDM) methods were combined in order to evaluate the HV according to factors that refer to floods and hydrological drought occurrences, hydrotechnical structure coverage, the drainage network, land use, and landforms characteristics. The Analytic Hierarchy Process (AHP) was applied to weigh these factors and the resulting relative importance values were integrated in the Technique for Order of Preference by Similarity to Ideal Solution (TOPSIS), by which the alternatives represented by the catchments were ranked. The attenuation of the HV through time follows an East–West direction, from the lower elevations of the Moldavian Plateau to the heights of the Carpathians. Hydrological droughts are more likely to occur in the Eastern part of the analyzed territory, while the western section displays a certain propensity for floods. The results may be used by local and national authorities in order to improve the hydrological risk mitigation strategies, and to develop more targeted water management projects, properly calibrated to the conditions of the Northeast Development Region in Romania.

**Keywords:** hydrological vulnerability; AHP; TOPSIS; multi-criteria assessment; NE Romania

## 1. Introduction

Vulnerability covers many definitions, the classical one referring to the potential loss that particular hazards may cause to the elements at risk [1]. Vulnerability, together with the associated hazard, is a risk component that expresses the susceptibility to harm, the damage level that results after the occurrence of a destructive event. More disaster studies place vulnerability at the core of the problem and emphasize the necessity of modifying it in order to ensure survival and development [2,3]. This multi-faceted concept has been studied using various approaches, leading to a fragmented research pattern. Its multiple definitions may be explained by the necessity to integrate it in specific contexts and also by the ontological and methodological particularities of the scientific disciplines that use it [4].

The hydrological vulnerability (HV) depends on the response of the natural and anthropic components that define a certain geographical area to the impact of extreme hydrological phenomena [5]. Frequently, this mix of natural and socioeconomic conditions,

together with hydrotechnical structures and other engineering infrastructure elements, determine the amplification of the aforementioned response, resulting in more powerful effects of floods and hydrological droughts. Depending on the occurrence time and area, the length of manifestation, propagation mode, and the aftermath of dangerous hydrological phenomena, the HV displays different spatial and temporal aspects according to the vulnerabilities specific to other systems. These vulnerabilities may be directly or indirectly associated with hydrological phenomena or with the social and cultural features of the exposed human communities [6,7]. As the effects of the hydrological phenomena affect human communities in more significant ways, and the perception of those exposed to them sharpens, HV assessments become more comprehensive and effective. In this context, these assessments are the prerequisites of the decisional processes regarding land use and territorial planning.

Vulnerability may be analyzed using various indicators, their suitability to express either the causal relationships that changes are based on, or the consequences of change, being key elements. As in the cases of vulnerability definitions, models, or evaluation methodologies, there is no consensus regarding the suitability, specificity level, or aggregation of vulnerability indicators [8,9].

Oftentimes, the need to integrate a large number of indicators that may be conflicting emerges, which calls for a multi-criteria approach. In this context, multi-criteria decision making (MCDM) methods prove to be effective because they integrate contrasting indicators, fulfilling the role of the evaluation criteria that the vulnerability assessment relies on. Moreover, Ref. [5] highlights the difficulties of measuring HV and argues for a case-by-case, repeated assessment approach. These requirements are easy to fulfil using MCDM methods, due to the fact that they are adaptable and intuitive.

This paper assesses the HV of 81 drainage basins in the northeast of Romania using two MCDM methods, for three periods: 1990–1999, 2000–2009, and 2010–2018. The underlying hypotheses are (i) the HV may be estimated using the proposed MCDM framework and (ii) the levels of HV change in time. Hydrological, land use, and constant indicators were used as criteria/factors relevant for the evaluation of the HV via the Analytic Hierarchy Process (AHP). The weights established by this method were integrated into the three assessments specific to each period taken into account. By replicating the relationships between the factors across space and time, the criterion of consistency was fulfilled. The alternatives represented by the 81 watersheds were ranked using the Technique for Order of Preference by Similarity to Ideal Solution (TOPSIS). The resulting scores were examined in order to acquire a better understanding of the evolution of HV levels in the areas of interest, making use of GIS instruments. The combination of MCDM methods and GIS techniques has been extensively used in studies relating to site suitability [10], site evaluation [11], or other vulnerability assessments [12–18].

The novelty of the paper consists of (i) aggregating flood vulnerability and hydrological drought vulnerability as part of the overall HV of the study area, together with the factors that increase or decrease the susceptibility of occurrence of the named destructive hydrological phenomena, and (ii) using a combination of MCDM that has not been used before to assess HV, both in relation to the study area and in the scientific literature. Such MCDM vulnerability analyses are only emerging in the Romanian scientific literature, meaning that the paper broadens the range of HVassessments in this country, from both a methodological point of view and an area of interest perspective.

Nonetheless, the proposed framework aims to contribute to the field of HV analysis by bringing to light the necessity to integrate multiple types of vulnerability (i.e., flood vulnerability, hydrological drought vulnerability); an approach that is often dismissed (for the benefit of focusing on a single hazard vulnerability). This is of particular importance when considering climate changes and the subsequent alterations of hydrological processes and phenomena. The multi-temporal approach is also beneficial to the integration of such interactions in HV assessment, as climatic and hydrological modifications may not be observed when analyzed in short time frames. The methodological framework may be

replicated, focusing on other study areas, but it should be emphasized that the suitability of the indicators may be altered. In order to avoid this, the methodology needs to be applied focusing on catchments that present various elevations, slopes, hydrological network densities, and that are affected by both floods and hydrological droughts, depending on the season and exact location.

## 2. Study Area

The study area includes 81 drainage basins located in the NE of Romania, overlapping the administrative territories of Suceava, Botoșani, Neamț, and Iași counties. It extends over 25,759.5 km$^2$ and its average elevation is 476.8 m, while its average slope value is 8.4°. The western part of the study area is dominated by the high elevations of the Carpathians, which are separated from the lower Moldavian Plateau that lies in the east, by a transition area of hills and depressions (the Moldavian Subcarpathians) and by the depression area of the Siret Corridor (Figure 1).

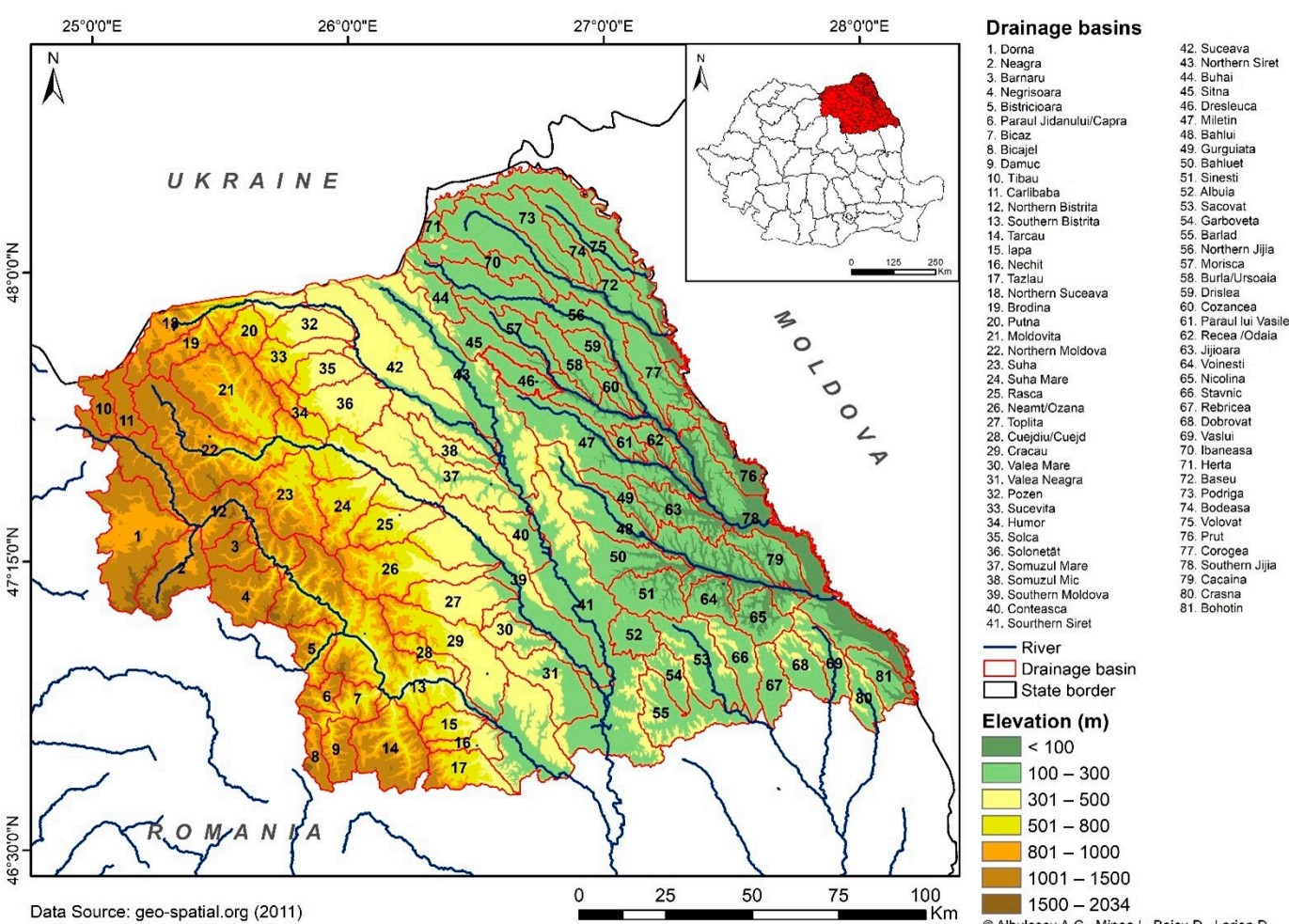

**Figure 1.** Location of the study area.

Most of the drainage basins have elongated shapes that follow the north west–south east tilting direction of the geological strata. The smallest watershed has an area of 41.3 km$^2$ (the Herța) and the largest one covers 1624.3 km$^2$ (the southern Bistrița), but more than half of the drainage basins extend over 100–300 km$^2$.

The main two collector rivers in the study area are the Siret River, flowing in the median depression section of the analyzed territory, and the Prut River, bordering it to the east. Both rivers are tributaries of the Danube and their confluence is located much further to the South of the study area. The main tributaries of the Siret descend either from the

Carpathian Mountains in the West (the Suceava, northern and southern Moldova, northern and southern Bistrița, and Bistricioara rivers) or join the collector from the left, crossing plateau regions (the Bârlad River and its tributaries). The main tributaries of the Prut River are shorter and cross the same plateau regions, joining the collector from the right (the Miletin, Bahlui, Jijia, Volovăț rivers).

### 3. Methodology

As any other type of vulnerability, HV cannot be directly measured, but estimated using factors that influence it and act as proxies with different suitability degrees. This paper proposes a methodological framework that uses a series of criteria and sub-criteria that converge to form the overall HV of the 81 catchments (Figure 1). To this end, the hydrological, land use, and constant factors function as sub-criteria, while the categories per se represent the evaluation criteria. The framework is based on two MCDM methods that fulfil different tasks: AHP is used in order to establish the relative importance of the evaluation sub-criteria (i.e., the contribution of each sub-criterion to the formation of the overall HV), while TOPSIS is applied to rank the alternatives (i.e., the considered catchments) according to their performance scores relating to the sub-criteria.

Firstly, we present the computation procedures of the data regarding the spatial units and the factors that influence their HV levels (Figure 2). The next part of the section represents a presentation of the MCDM methods that integrate these data. The next step consists of ordering the scores obtained via TOPSIS in a decreasing order, as the higher ones correspond to higher levels of HV. The HV levels are established using a geometric progression. The methodological framework is applied for each of the three periods and the results are comparatively analyzed.

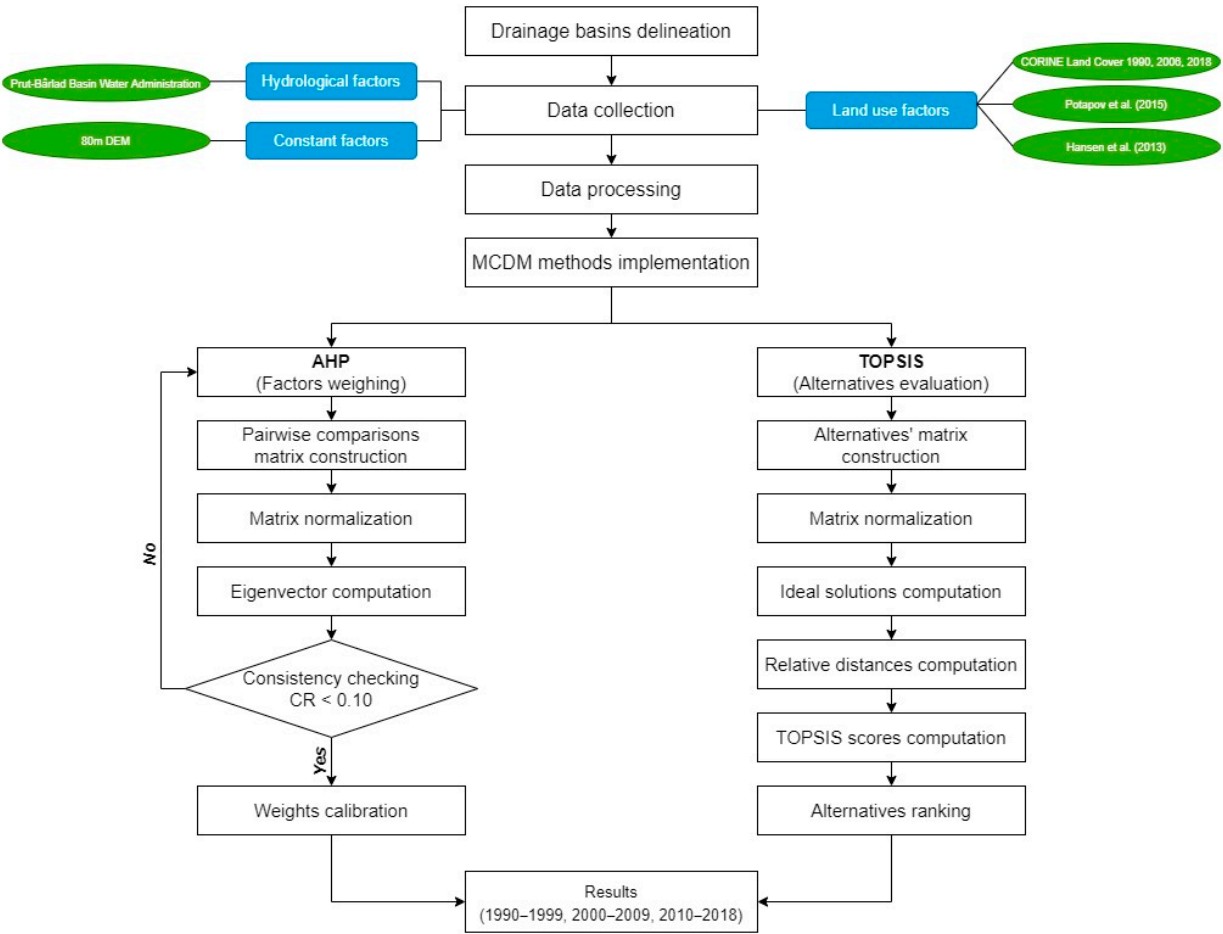

**Figure 2.** Methodological flowchart.

It should be noted that vulnerability analysis is just a component of the larger hydrological risk analysis, and that this study deviates from the norm by including two elements related to hazards into the HV (the incidence of floods and the intensity of hydrological droughts)—that would normally relate to the risk notion. This is motivated by the fact that the two aforementioned factors were not comprehensive enough to cover the hazard component of a broader hydrological risk assessment.

### 3.1. Delineation of the Drainage Basins

All drainage basins from the northeast part of Romania have a particular HV. Annually, several floods caused by torrential rainfall, and droughts generated by the lack of precipitation, occur. To delineate the drainage basins, a semiautomatic method is performed in ArcGIS, using the spatial database of the ASTER DEM dataset, of 25 m, and the Strahler method of hierarchizing hydrographic basins [19]. The basins are selected at a threshold of 10,000 cells, in order to correlate the results with the hydrological observations made by the national monitoring system. Applying this method, 81 Strahler-based fourth order basins result, each having a catchment area with no less than 40 km$^2$ [20].

### 3.2. Dataset

This assessment considers three categories of criteria/factors, each containing three factors that work as indicators for hydrological, land use, and constant aspects that influence the HV of the 81 drainage basins. Depending on their influence on HV, the sub-criteria may work as benefit elements when high values determine high HV levels, or as non-benefit sub-criteria when the opposite judgement is valid (Figure 3). The aforementioned factor categories are assigned weights that reflect the impact they have on HV: 50% for the hydrological factors, 30% for the land use factors, and 20% for the constant factors. The factors are evaluated via AHP at a category level and their final weight is calibrated with the weight of the specific category by multiplication.

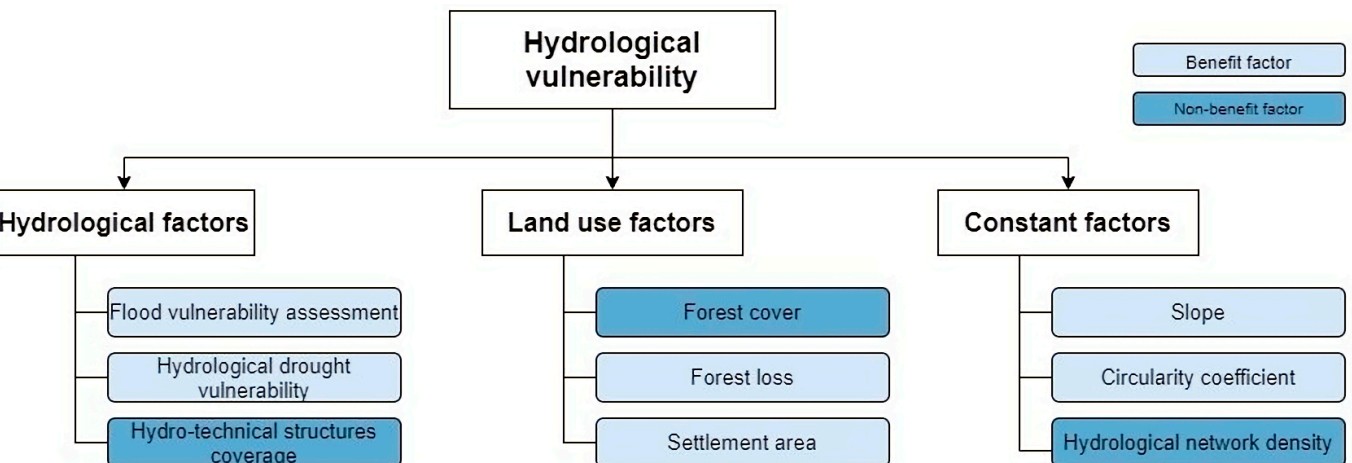

**Figure 3.** The hydrological, land use, and constant factors. Integration of the relations established between the considered factors is addressed by attributing category level relative importance values. The influence of other factors on the hydrological ones is different for each of the 81 drainage basins. For example, there are catchments where the forest covers are extensive, and influence on the occurrence of floods are significant, but there are catchments with small forest areas, where the influence of this factor is neglectable. The same can be stated for the other land use and constant factors. This explains why the relations between the factors are not key points of the proposed framework and why a weighing MCDM method that does not consider this issue, namely the AHP, is applied instead of a method based on criteria connections, such as the Analytic Network Process (ANP).

A reminder is in order here: floods and hydrological droughts are the main hydrological phenomena analyzed in order to evaluate the HV of a region or a drainage basin. The dynamics of these phenomena [21], together with one of the climatic parameters [22], are the basis of decisions regarding water resource management [23]. Therefore, the hydrological factors have a relative importance of 50% in the present assessment.

### 3.2.1. Hydrological Factors

Hydrological data for the period of 1990–2018 were acquired from the Prut-Bârlad Basin Water Administration Branch managed by the Romanian Waters National Company. Monthly data regarding the frequency of floods and discharge values were used in order to compute the hydrological factors. For the drainage basins where no hydrological dataset was available, data from the neighboring basins with similar geographical conditions were used. An overview of the values of these indicators in the study area is presented in Figure 4.

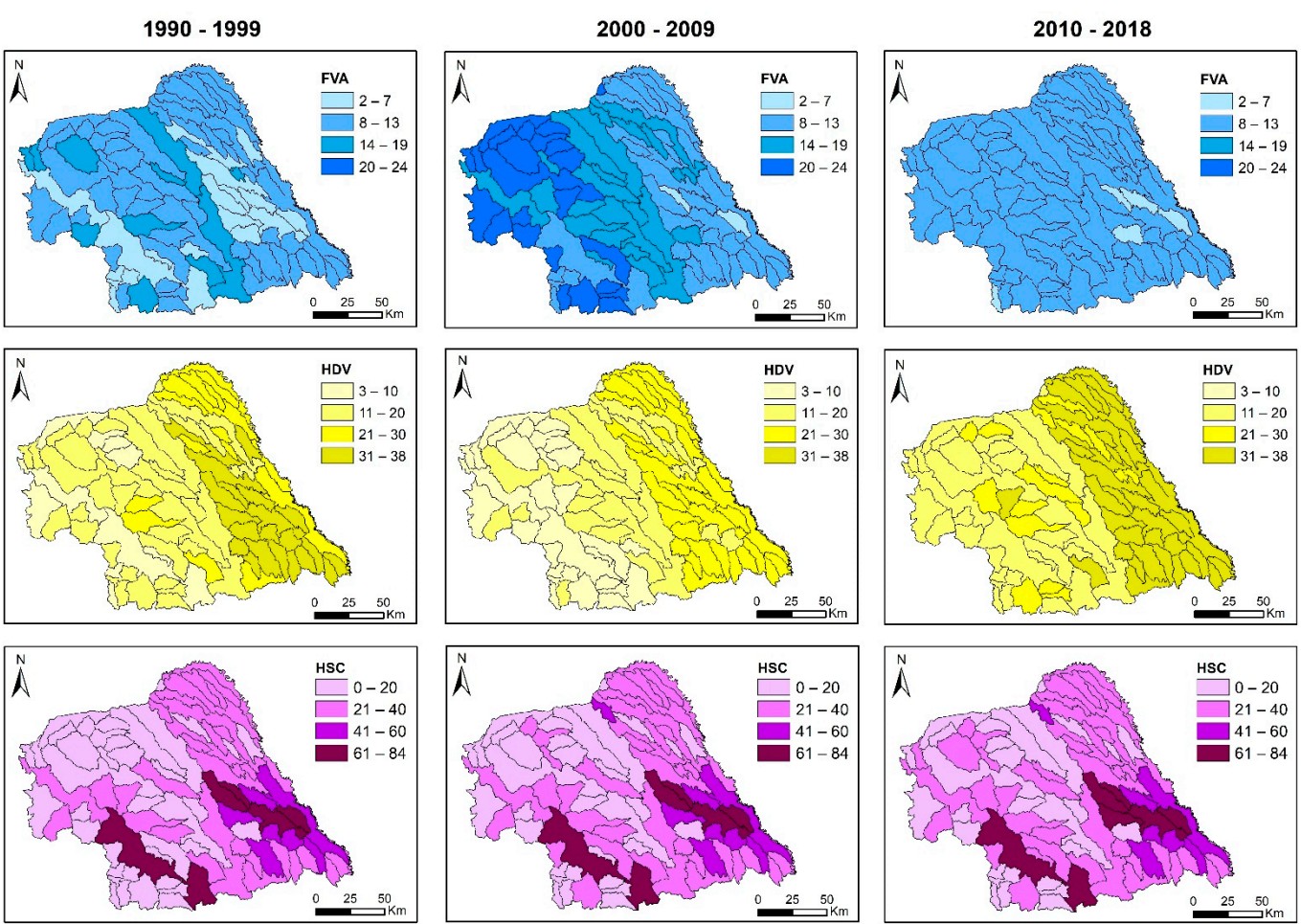

**Figure 4.** The evolution of the hydrological factors' values in the study area.

Flood vulnerability assessment (FVA) was computed based on data regarding the monthly high flows produced at each hydrometric station. Each monthly high flow value was compared with the value corresponding to the flood rate and the exceeding values were taken into account in the HV analysis [24]. According to this evaluation, the drainage basins were assigned the total number of floods higher than the flood rate for each decade.

Hydrological drought vulnerability (HDV) was processed according to the [25] streamflow drought index (SDI). Hydrological drought periods can be investigated using the monthly streamflow values for each hydrometric station and the methods associated with

the standardized precipitation index (SPI, described by [26]). The Kolmogorov–Smirnov (K–S) test for the 0.05 significance level was applied in order to verify the suitability of the streamflow values, while the log-normal distribution was used to decrease the skewness values of all streamflow series [27]. The results are classified as follows: SDI values under −2 correspond to extreme drought, the SDI values between −2 and −1.5 indicate severe drought, SDI values between −1.5 and −1 correspond to moderate drought, values of SDI between −1 and 0 suggest minor drought, and SDI values over 0 show the absence of droughts. Each month was analyzed according to this classification and the months with extreme, severe, and moderate hydrological droughts are summed for each hydrometric station.

Hydrotechnical structure coverage (HSC) was obtained by dividing the area covered by hydrotechnical structures (reservoirs, polders, deviation channels, levees, etc.) by the total area of the drainage basin [28]. In several cases, the hydrotechnical structures serve multiple purposes, among which, regulating the streamflow and attenuating flood waves are the most common. They also reduce the impact of extreme hydrological phenomena, such as floods and hydrological droughts on human communities. The paradigm implies that this type of structure rarely covers the whole drainage basin [29].

### 3.2.2. Land Use Factors

These factors relate to the impacts of different land use on the hydrological dynamics of the drainage basins. Forests fulfil various ecosystem services, of particular importance in this context—the regulation and provisional ones [30,31]. The influence of forest areas over flood hazards has been largely studied, and their moderator role on hydrological and climatic processes has been reiterated [19,32,33]. Larger forest areas imply that the impact of dangerous hydrological phenomena is delayed and attenuated, as tree canopies and radicular systems regulate streamflow and absorb part of the rainfall.

In addition, the effects of the settlement area particularities on the hydrological dynamics of the drainage basins are of vital importance [34,35]. Settlement areas are usually covered by hardened concrete surfaces, which preclude water infiltration. The same areas may have sewage systems that significantly disrupt infiltration and streamflow patterns. Figure 5 illustrates the evolution of these factors in 1900–2018.

Forest cover (FCOV) was obtained by processing CORINE Land Cover data specific for the three periods: 1990 data for the 1990–1999 period, 2006 data for 2000–2010, and 2018 data for 2010–2018. The factor integrates areas of broad-leaved forests, coniferous forests, mixed forests, transitional-wood shrubs, and agro-forestry areas, accounting for patches of forest vegetation larger than 25 hectares. The forest cover is expressed as the percentage of the forest area reported to the total watershed area. For most of the drainage basins, the forest areas represent less than half of their territories, and only eight of them have forests that cover more than 80% of the total area.

Forest loss (FLOSS) was computed using LANDSAT data processed from (i) [36], for the years 1989–2000 corresponding to the 1990–1999 period, 2001–2006 combined with 2007–2012 for the 2000–2009 period, and (ii) [37] 2013–2018 for the time period of 2010–2018. This indicator is expressed as a percentage of the forest loss registered within a drainage basin in a given period, reported to the total area of that basin. More than half of the drainage basins in the study area register forest loss that account for less than 1% of their area and only fifteen have forest loss levels higher than 5%. The period 2000–2009 had the highest levels of forest loss in the study area.

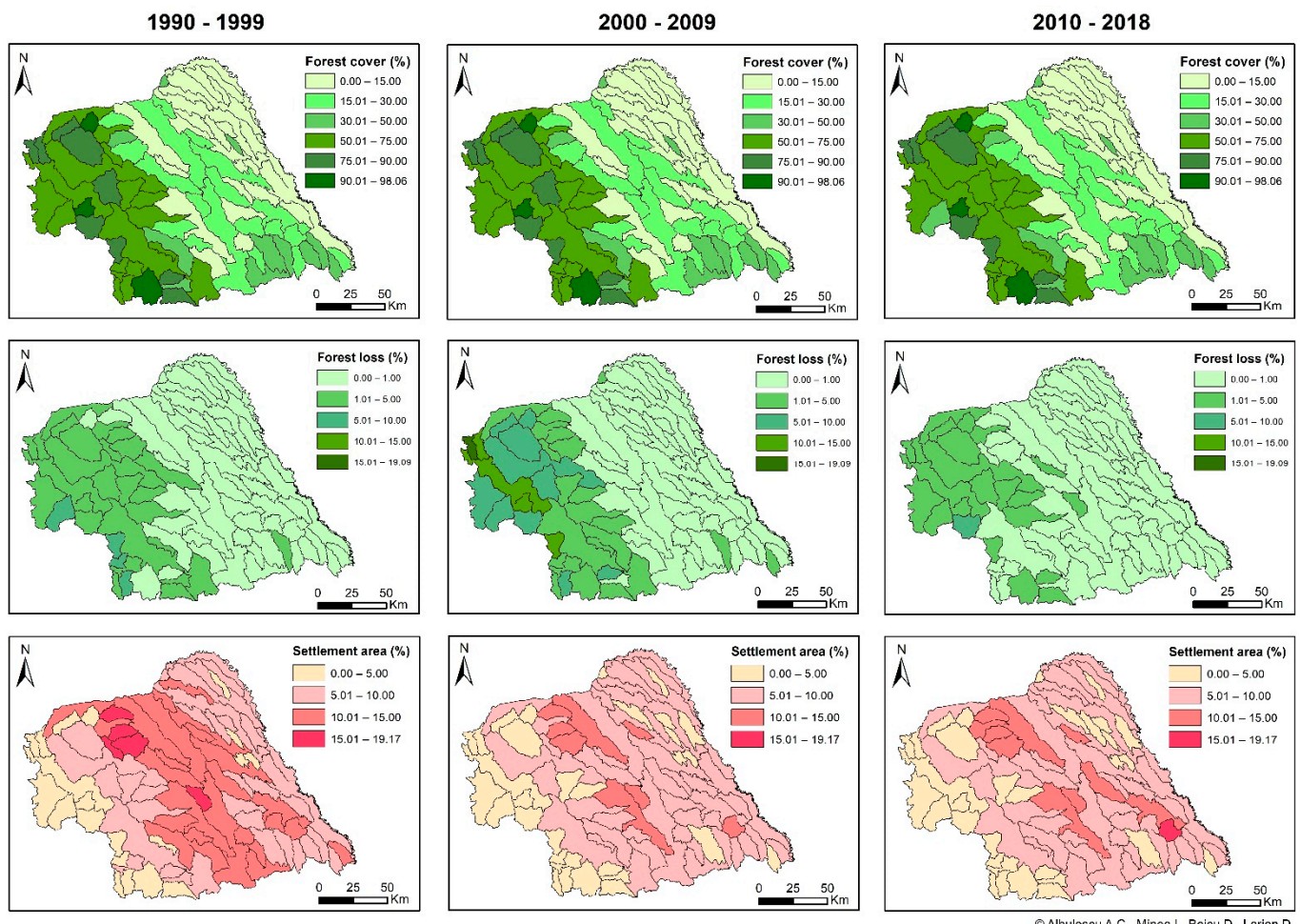

**Figure 5.** The evolution of the land use factor values in the study area.

The settlement area (SETTLE) was computed as the percentage occupied by continuous urban fabric and discontinuous urban fabric in the drainage basins. The rural settlement areas were left out, as the coverage of impermeable surfaces or sewage systems within their territories are insignificant, producing no (or little) modification on the dynamics of the streamflow. The same CORINE Land Cover data periods (1990, 2006, 2018) are used for the computation of this indicators' values. Only 29 of the 81 drainage basins have 10–19.17% settlement areas within their boundaries, and in 42 of them, the settlement areas cover less than 6%.

### 3.2.3. Constant Factors

The constant factors category includes those indicators that directly and indirectly influence the evolution of hydrological phenomena that impose risks, and that slightly change over time. The slope, drainage density, and the circularity coefficient of drainage basins tend to remain constant, as the processes that lead to their modifications usually require extended periods of time, and only in particular cases do they act in forceful and definite manners. The same values of these indicators are used for computations for all of the considered periods (Figure 6), which also explains the 20% relative weight of the category.

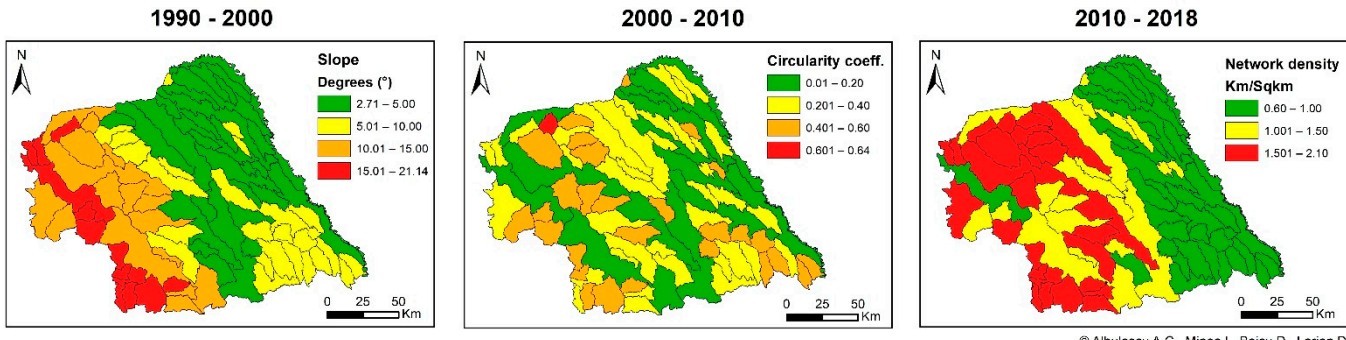

**Figure 6.** An overview of the constant factor values in the study area.

The geology of the study area is also worth considering, but such factors are not included in the assessment. The geological structure of the eastern part of Romania generated a series of specific development conditions of the drainage basins. The East-Carpathian Crystalline-Mesozoic Zone [38], with hard rocks of low permeability, contributed to the formation of elongated catchments and developed along fault lines. In the platform area, towards the northeast part of Romania, the Sarmatian and Quaternary geological deposits, made of clay, marls, and sands with significant thickness, generated extensive drainage basins [39,40].

Slope (SLOPE) is a static factor that influences HV because it is directly related to the relief. Its values are automatically generated using an ASTER DEM model of 80 m; 36% of the drainage basins are characterized by low slope values (0–5°) that favor slow/diminished streamflow along the sides of the drainage basins and allow for water deposition in the small depressions of the topographic surface or in floodplains. The drainage basins with slope values between 5° and 10° represent 29% of the total number of watersheds, while the ones with slopes of 10–15° represent 18% of it. Slope values higher than 15° also account for 18% of the drainage basins, being associated with significant streamflow generated by rainfall of snowmelt and with low infiltration rates [41].

Hydrological network density (NETDENS) represents the total length of the river channel that is permanently covered by streamflow divided by the area of the drainage basin. It influences the streamflow volume, as higher values favor the occurrence of floods, with a negative impact on the environment [42]. Moreover, lower values of this factor are correlated with hydrological droughts [43], which are more frequent in the drainage basins located in the eastern part of the study area.

Circularity ratio (CR) ranks among the morphometric parameters that condition the occurrence and the evolution of hydrological phenomena that constitute threats to the environment and human communities. It is computed using Miller's formula [44] using the area and the perimeter of the drainage basins. Its higher subunitary values indicate a more rounded shape of the drainage basin at hand, implying that it is more subject to risk related to high streamflow [45].

### 3.3. Multi-Criteria Decision Making Methods

Multi-criteria decision making (MCDM) methods are used to solve problems with discrete decision spaces, meaning that the set of alternatives that contains the solution or that needs to be ranked is predefined [46]. The AHP and TOPSIS serve different purposes in the proposed framework, the first one being used to determine the factor weights and the second one to rank the alternatives based on their suitability to each criterion (Figure 2). Both methods were applied to each of the three periods, with the weights of the AHP being maintained and the TOPSIS scores varying according to the evolution of the alternative values in relation to the factors.

By combining these two MCDM methods, two main aspects of the assessment in question were addressed: the task of establishing the relative importance of the factors at a category level was fulfilled by AHP and the necessity to rank a large number of alternatives

was solved by implementing TOPSIS. While another factor weighing MCDM methods could have been used, AHP was preferred because the proposed framework does not take into consideration the relations between the aforementioned factors—a situation for which ANP would have been adequate, but uses a structure of main criteria and sub-criteria and integrates, and a category level pairwise comparisons approach. Moreover, TOPSIS is a robust ranking method that can evaluate alternatives with performance scores that are expressed using different measuring units. This combination allows for a salient MCDM assessment of the HV in the area of interest.

The weighing of the factors is based on the judgments of a team of four experts knowledgeable on hydrology, climatology, land cover, and forest dynamics. They were grouped according to their expertise fields, and each was asked to perform the pairwise technique on the set of indicators. Their combined expertise ensured a proper assessment of the contributions of the hydrological, land use, and constant factors to the overall hydrological vulnerability of the study area. The judgments were checked for consistency, in order to avoid erroneous implementations of the MCDM methods. The final weights were obtained by comparing the results, with each expert supporting his/her opinion with on-point arguments. This discussion forum motivated the final weights of the factors, also serving as a consistency-checking instrument.

### 3.3.1. AHP

AHP is a hierarchy-based MCDM method that emerged in the 1970s [47] and became one of the most popular due to its flexible and intuitive framework. It decomposes the decision-making problem by structuring its components on different levels: the first one refers to the goal, the next one to the evaluation criteria that are relevant to the goal, and the last level to the alternatives that need to be evaluated considering the criteria.

The factors on the same level are compared in pairs, using a scale of predefined values and their reciprocals. These are organized in a matrix of $m \times m$ elements that are used to compute the eigenvector that contains the relative importance values of the elements. The pairwise comparison matrix is normalized (1) and the arithmetic average operation is applied to each line in order to obtain the eigenvector (2) [47,48].

$$NM = \frac{M_{ij}}{\sum_{l=1}^{m} M_{lj}} \tag{1}$$

$$W_i = \frac{\sum_{l=1}^{m} NM_{il}}{m} \tag{2}$$

where $M$ is the original pairwise comparison matrix, $NM$ is the normalized version of the pairwise comparison matrix, $m$ is the number of elements, $W$ is the eigenvector.

The goal of the problem at hand is to assess the HV of the watersheds in the study area. On the second level, we find the criteria represented by the categories of hydrological, land use, and constant factors, followed by the third level that comprises the factors included in each of the aforementioned categories. These factors function as evaluation sub-criteria that are weighed via AHP.

In this paper, AHP was used only to evaluate the factors because the set of alternatives was too large and the 81 drainage basins needed to be ranked using a different MCDCM method, namely TOPSIS. Because each category contains only three factors, the set of predefined values that expresses the relations between the factors was reduced to a simpler form: 1—the two factors have equal importance, 1.5—one factor is slightly more important than the other, 2—one factor is more important than the other. This simplification deviates from the standard AHP, but is more suitable for the problem at hand, because the full scale of 1–9 values used for pairwise comparisons would create significant discrepancies between the relative importance values of the factors. Therefore, a three-point scale offers more balanced weights for the three triplets of factors.

The judgements used in the pairwise comparison process were elaborated by the team of experts, considering information from the scientific literature, as well as their expertise concerning hydrological risks, vulnerability, and the study area. Tables 1–3 contain the pairwise comparison matrices for each category of factors.

**Table 1.** The pairwise comparison matrix of the hydrological factors.

|  | FVA | HDV | HSC |
|---|---|---|---|
| FVA | 1 | 1 | 2 |
| HDV | 1 | 1 | 2 |
| HSC | $\frac{1}{2}$ | 1/2 | 1 |

**Table 2.** The pairwise comparison matrix of the land use factors.

|  | FCOV | FLOSS | SETTLE |
|---|---|---|---|
| FCOV | 1 | 1/2 | 1 |
| FLOSS | 2 | 1 | 2 |
| SETTLE | 1 | 1/2 | 1 |

**Table 3.** The pairwise comparison matrix of the constant factors.

|  | SLOPE | CR | NETDENS |
|---|---|---|---|
| SLOPE | 1 | 1/1.5 | 1/1.5 |
| CR | 1.5 | 1 | 1 |
| NETDENS | 1.5 | 1 | 1 |

Moreover, AHP includes a consistency checking algorithm that ensures the validation of the judgements. The eigenvalue of the eigenvector is computed (3) in order to obtain the consistency index (4) that is further divided by a predefined random consistency index, dependent on the number of elements. The result represents the consistency ratio and it has to be lower than 0.1 for a valid judgement (5) [49].

$$x = \sum_{i=1}^{m} W_i \cdot \sum_{j=1}^{m} M_{ji} \tag{3}$$

$$CI = \frac{x - m}{m - 1} \tag{4}$$

$$CR = \frac{CI}{RI} \tag{5}$$

where: $x$ is the eigenvalue, $CI$ is the consistency index, $RI$ is the random consistency index, and $CR$ is the consistency ratio.

### 3.3.2. TOPSIS

TOPSIS was developed by [50] as a MCDM method that may be used to rank a large number of alternatives according to their distances to the positive and ideal solutions. The positive ideal solution is defined as an optimal solution, while the negative ideal solution is considered the least favorable. Thus, the method determines the compromise solution as the one closest to the positive ideal solution and furthest from the negative ideal solution, in terms of Euclidean distance [51].

In this case, the alternatives are represented by the 81 catchments in NE Romania. The positive ideal solution represents the catchment with the highest HV, while the negative ideal solution represents the catchment with the lowest HV. Thus, TOPSIS places each catchment somewhere between these two extremes, based on the specific values of each of the considered factors.

The method is suitable for ranking alternatives that are measured using different units, which adds to its adaptability. The values of the alternatives regarding different criteria are organized in a matrix that has to be normalized (6). Further on, the elements of the normalized matrix are multiplied by the factors' weights (7) [52].

$$NX_{ij} = \frac{X_{ij}}{\sqrt{\sum_{i=1}^{m} X_{ij}^2}} \tag{6}$$

$$Y_{ij} = W_j \cdot NM_{ij} \tag{7}$$

where $X$ is the original matrix with the values of the alternatives, $NX$ is the normalized matrix, $m$ is the number of factors, $W$ is the vector containing the factors' weights, and $Y$ is the matrix that contains the weighed scores of the alternatives in relation to the criteria.

Based on these, the positive and negative ideal solutions are computed for each of the considered factors ((8) and (9)). The next steps consist of computing the distances ($d^+$, $d^-$) of each alternative to the positive, respectively, the negative ideal solutions ((10) and (11)). Finally, the relative distances ($D^+$) of the points represented by the alternatives respecting the ideal solutions are computed (12) and sorted from the largest to the lowest [52].

$$Y_j^+ = \begin{cases} \max_i Y_{ij} \ for \ j = 1, \dots, K \\ \min_i Y_{ij} \ for \ j = K+1, \dots, m \end{cases} \tag{8}$$

$$Y_j^- = \begin{cases} \min_i A_{ij} \ for \ j = 1, \dots, K \\ \max_i A_{ij} \ for \ j = K+1, \dots, m \end{cases} \tag{9}$$

$$d_i^+ = \sqrt{\sum_{j=1}^{m} \left( Y_{ij} - Y_j^+ \right)^2} \tag{10}$$

$$d_i^- = \sqrt{\sum_{j=1}^{m} \left( Y_{ij} - Y_j^- \right)^2} \tag{11}$$

$$D_i^+ = \frac{d_i^-}{d_i^+ + d_i^-} \tag{12}$$

where: the first $K$ factors are benefit factors and the rest are non-benefit factors, $Y^+$ is the positive ideal solution, and $Y^-$ is the negative ideal solution.

The alternative with the highest TOPSIS score has the highest HV because it is located the closest to the positive ideal solution and the farthest from the negative ideal solution. The TOPSIS scores are classified according to geometric progression, into five classes of HV. This progression is preferred to the others because it offers the most balanced distributions of the 81 drainage basins, allowing for better insights regarding the evolution of the HV level of the drainage basins over the three periods of time.

## 4. Results

By calibrating the weights obtained via AHP at the category level with the overall weight of the specific category, two hydrological factors emerge as the most important, being followed by a land use one (Table 4, Figure 7). FVA and HDV each have a 20% relative importance because they express occurrence related aspects of the hydrological phenomena that directly affect the drainage basins. The forest loss factor is also very important, obtaining a 15% relative importance, due to the destabilization effect of tree felling on the hydrodynamics of the river network. The next factor in order of importance is HSV (10%), which shows to which extent a particular drainage basin is protected by the negative effects of hydrological hazards by hydrotechnical structures.

**Table 4.** The category and overall weights of the hydrological, land use, and constant factors.

| | Weights at Category Level | Weights at Category Level (%) | Final Weight | Final Weight (%) |
|---|---|---|---|---|
| **Hydrological factors** | **0.5** | **50** | | |
| FVA | 0.40 | 40 | 0.20 | 20 |
| HDV | 0.40 | 40 | 0.20 | 20 |
| HSC | 0.20 | 20 | 0.10 | 10 |
| **Land use factors** | **0.3** | **30** | | |
| FCOV | 0.25 | 25 | 0.075 | 7.5 |
| FLOSS | 0.50 | 50 | 0.15 | 15 |
| SETTLE | 0.25 | 25 | 0.075 | 7.5 |
| **Constant factors** | **0.2** | **20** | | |
| CR | 0.25 | 25 | 0.075 | 7.5 |
| NETDENS | 0.25 | 25 | 0.075 | 7.5 |
| SLOPE | 0.25 | 25 | 0.05 | 5 |

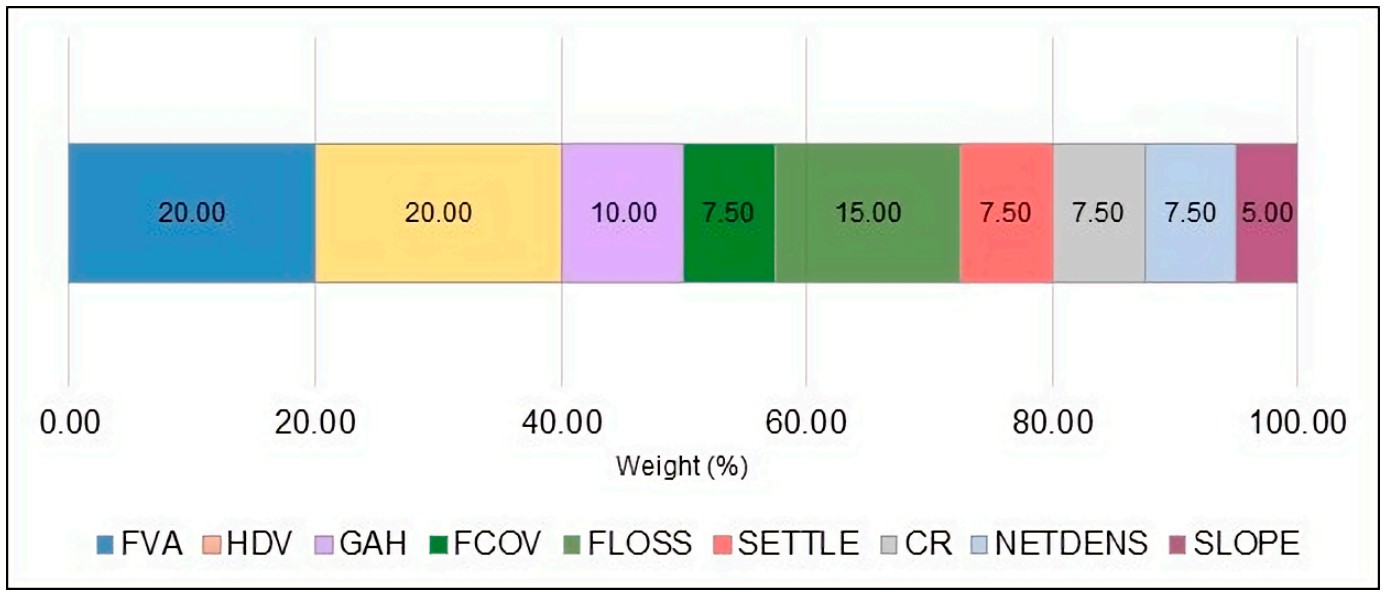

**Figure 7.** The weights of the hydrological, land use, and constant factors.

Forest cover and settlement area have equal relative importance values (7.5%), because the dominance of one factor over the other is hard to establish and varies from one drainage basin to another. The circularity coefficient and the network drainage density rank the same, with 7.5% weight, because the former directly conditions the particularities of flood occurrence and the latter influences the probability of occurrence of hydrological drought (Figure 7). Since the hydrological factors that relate to those hazards are thought to be equally important, these two constant factors that relate to them are placed on the same importance level. Slope has the lowest weight of all factors (5%), due to its little spatial variance and to the fact that it relates more to flood occurrence and influences droughts to a significantly less extent.

Most of the drainage basins have medium HV levels in 1990–2009 and low HV in 2010–2018. The extreme classes follow opposite trends across time: the number of watersheds with very low HV increases from the beginning (one element) to the end (five elements) of the reference period, while the number of drainage basins with very high HV decreases from seven to only one. The same increasing tendency is registered by the category of low HV. On the other hand, the drainage basins with high HV levels became more numerous in 2000–2009 (10 elements) compared to 1990–1999, and then they reduced by 50% in 2010–2018 (Table 5).

**Table 5.** The HV classes and their assigned number of drainage basins in 1990–2018.

| HV Level | TOPSIS Scores | | No. of Drainage Basins | | |
|---|---|---|---|---|---|
| | Inferior Limit | Superior Limit | 1990–1999 | 2000–2009 | 2010–2018 |
| Very Low | 0.2098 | 0.2705 | 1 | 1 | 5 |
| Low | 0.2706 | 0.3487 | 11 | 31 | 57 |
| Medium | 0.3488 | 0.4496 | 55 | 36 | 13 |
| High | 0.4497 | 0.5796 | 7 | 10 | 5 |
| Very high | 0.5797 | 0.7473 | 7 | 3 | 1 |

This means that the overall HV in the study area has moderation tendencies that advance on an east–west direction, from the lower areas of the Moldavian Plateau to the higher hills and mountain ranges that lie to the west (Figure 8). For all periods, the areas with high and very high HV are located in the northwest of the study area. This is explained by the convergence of high elevation, slope, network density, circularity, and rainfall values, which favors more frequent floods, and of low or very low HSC values.

*4.1. Hydrological Vulnerability between 1990 and 1999*

This period was the one with the highest number of drainage basins with very high HV (the Rasca, Humor, Cârlibaba, Capra, Neagra, Bistricioara, and Dămuc catchments) and one of those when the very low HV elements reached a minimum (the catchment of Southern Bistrița). The better half of the watersheds had medium levels of HV and more drainage basins with low HV (11) than with high HV (7). The mountainous area in the west includes the drainage basins with high and very high HV, given the fact that floods tend to occur more often due to higher rainfall. Moreover, the Siret Corridor registers medium levels of HV in 1990–1999 due to significant FVA and HDV values, combined with the ones of the settlement areas, while the drainage basins with low HV lie along the Jijia and the Miletin rivers.

*4.2. Hydrological Vulnerability between 2000 and 2009*

In this period, the number of medium HV drainage basins reduced to 36, most of them acquiring a low level of the parameter. Nevertheless, the number of watersheds with high HV increased with three elements, the new ones being the catchments of the Iapa, Brodina, Neagra, Dorna, Northern Bistrița, Dămuc, and Bârnaru rivers, and the old ones the Moldovița, Negrișoara, and Northern Moldova catchments. Only three drainage basins have very high HV levels, namely the ones of Bistricioara, Cârlibaba, and Țibău rivers, whereas the southern Bistrița catchment maintains a very low level of HV. In the Moldavian Plateau, the low HV areas extend to the west, a direction that is also followed by the medium HV areas that replace the former high or even very high HV levels.

*4.3. Hydrological Vulnerability between 2010 and 2018*

The last period is the most moderate one, with only one drainage basin with a very high HV (the Negrișoara) and the highest numbers of units with very low (the southern Bistrița, Buhai, Bahlueț, Gurguiata, Cacaina catchments) and low HV, respectively (57 drainage basins). Most of these numerous low HV units are the former medium level drainage basins. Moreover, the number of catchments with high HV (the Bârnaru, northern Bistrița, Țibău, Putna, and Brodina catchments) decreased by half and kept their location in the northwest of the study area. In this period, the moderation tendency clearly extended to the west and northwest, from the rivers in the Prut catchment to the ones on the left side of the Siret River and those on its right side. This east–west trajectory follows the increase of elevation, from the Moldavian Plateau and the Siret Corridor towards the Subcarpathians and the Carpathians.

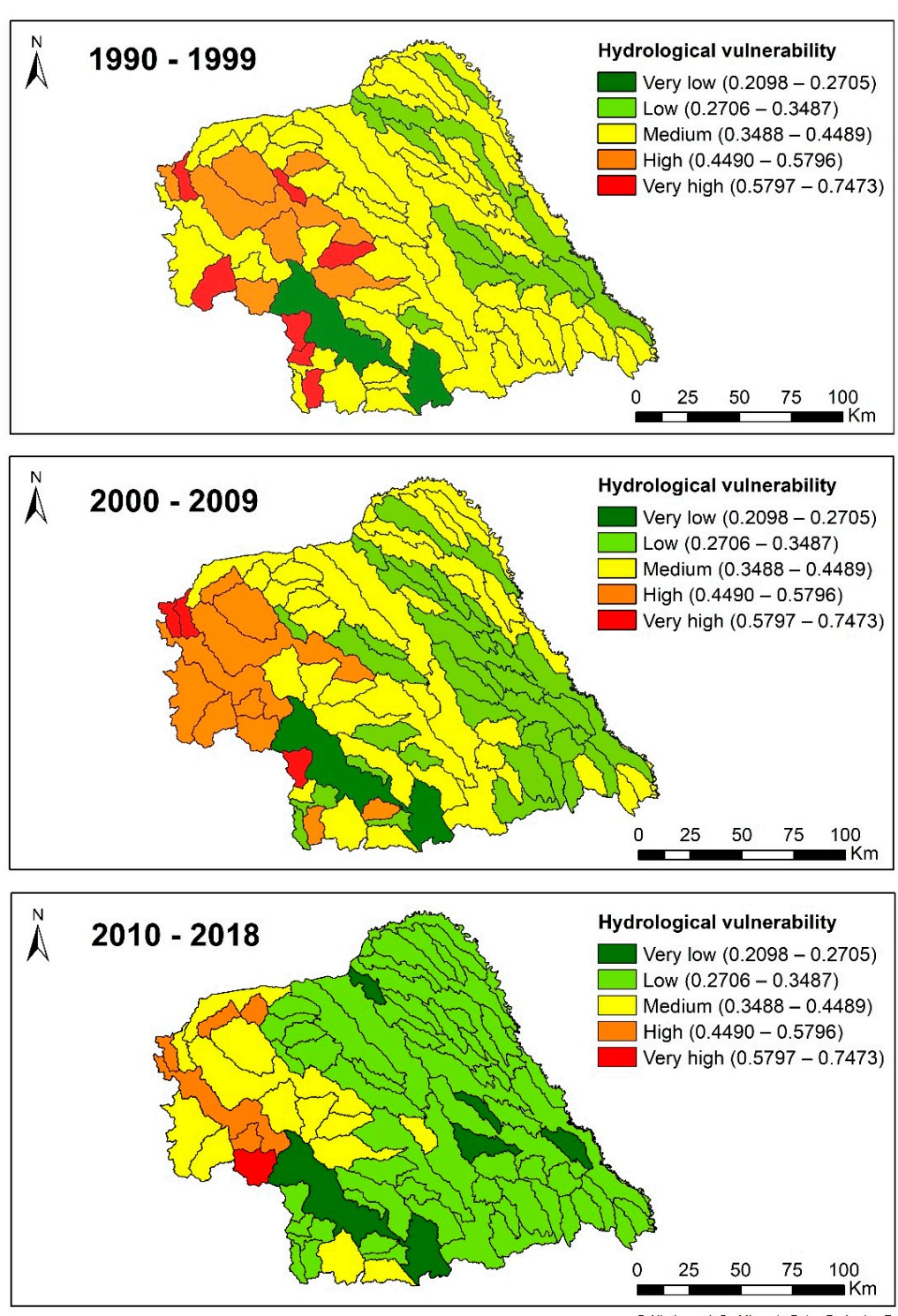

© Albulescu A.C., Minea I., Boicu D., Larion D.

**Figure 8.** The evolution of the hydrological vulnerability level in the study area in 1990–2018.

## 5. Discussion

Assessing HV through a multi-criteria approach is still an element of debate in the field of hydrological risk research. Usually, HV evaluations rely on GIS-based models that take into account drainage, rainfall, geomorphological and water extent aspects [6], or hydrological classification techniques [53]. There is a tendency to assess the vulnerabilities induced by particular hydrological hazards—most frequently floods [54–58], rather than aggregating the vulnerabilities induced by different hydrological hazards. Some of the

case studies on flood vulnerability use risk indexes [55], while others are based on flood vulnerability indices that vary depending on spatial scales [58–61], on parametric and physically modeling techniques [62], or GIS and remote sensing [56].

The presented methodological framework relates to two contrasting hazards (i.e., floods and hydrological droughts), which do not overlap in terms of time and space. This ensures a broader perspective on the hazards that threaten both the environment and the human communities in the study area. The considered watersheds are affected by both phenomena, due to the fact that they vary in terms of geological and geomorphological settings, and climatic particularities. Thus, the integration of two types of vulnerabilities allows for a more salient HV assessment.

Moreover, the assessment considers HV proxies related to land cover. This multi-faceted approach motivates the selection of MCDM methods as operationalization tools. Their implementation is hardly a novelty, as the AHP was used to assess the environmental vulnerability of drainage basins, considering factors related to rainfall, vegetation, soils, landslides, land use, and population [63]. In the field of multi-criteria environmental vulnerability research, Ref. [64] compares the weighted method (AHP) with the compromise one (VIKOR). Further on, a series of MCDM-based flood vulnerability assessments may be mentioned. Fuzzy TOPSIS served as an evaluation instrument in a multi-criteria flood vulnerability assessment, together with the Delphi technique [65], while AHP and GIS techniques were combined to assess urban flood vulnerability in Eldoret municipality, Kenya [66]. Other fuzzy versions of MCDM methods were implemented to detect flash flood areas [67,68] or to map urban flood vulnerability [69], in conjunction with GIS techniques. The scientific literature also provides examples of how MCDM methods and GIS are integrated to map out flood risks [70–72] or flood hazards [73–75]. However, the reviews on HV [76] and hydrological drought [77] assessments do not list such methodologies among common research practices.

This case study on the drainage basins in NE Romania contributes to the application of MCDM methods in HV assessment, due to the fact that the combination of the two MCDM methods has not been used before. Their suitability is proven by the capacity of such methods to integrate a set of criteria that have different natures and contrasting influences on HV, and also by the fact that they allow the evaluation of a high number of alternatives. Moreover, by applying the framework on three periods, the case study meets the demands formulated by [5], namely that HV assessments should be performed frequently, due to the changes of the underlying natural and socioeconomic conditions specific to drainage basins. Another argument in favor of the multi-temporal approach relates to the fact that it allows for the integration of hydrological alterations induced by climate changes, which have drawn more interest in recent decades, from the research community and laymen. This makes the proposed framework suitable for implementation focusing on other study areas with similar features (various geomorphological settings and climatic characteristics), that confront both floods and hydrological droughts, depending on the season and location.

A secondary element of novelty is that the study area has not been assessed from this point of view before; the present assessment serves as one of the first instruments that may help the elaboration of better hydrological hazard mitigation planning. This means that the results of the assessments cannot be compared to other of similar types, as there are no such assessments targeting the considered study area or integrating the two MCDM methods simultaneously. Thus, the results may contribute to the development of water management and hydrological risk reduction plans that are adjusted to the particularities of the Northeast Development Region of Romania. Ideally, the "traffic-lights" HV maps (Figure 8) should serve as cartographic instruments that inform decision-making at local and regional scales. They allow for hydrological risk mitigation in advance, based on the decadal evolution of HV levels, in an area that is affected by hydrological hazards located at the ends of the water quantity spectrum. It should be highlighted that such valuable insights may be provided for other study areas in the world, following the implementation of the proposed framework.

The limitations of the HV assessment concern the methodological aspects, meaning that the relations between the factors are not fully integrated, but acknowledged by attributing different relative importance values to the categories of factors. Moreover, the geological factors are missing from the category of constant factors. Other hydrological parameters could have been used, such as the evolution of the maximum or minimum streamflow at the catchment level or morphometric parameters specific to the drainage basins. However, the floods and hydrological droughts were selected because they have significant and immediate impacts on the environment and human communities, and because they are frequent in the study area. Another limitation is represented by the fact that the hydrotechnical structure coverage, which is used to express the anthropic influence on the rivers' discharge, registers insignificant modifications in time, because of the lack of financial investments in this domain. A third limitation concerns the exclusion of social vulnerability indicators, which would provide useful ques regarding the attitudes, experiences, and preparedness level of the human communities that inhabit the considered catchments.

## 6. Conclusions

All drainage basins have particular levels of HV that change in time, following the dynamics of the underlying factors referring to hydrological phenomena, land use transformations, and the slow processes that shape the landforms. Human interventions in the form of hydrotechnical structures and the engineering infrastructure act as amelioration instruments, reducing the HV and contributing to the formation of a sense of security.

The HV in the study area registered a moderation trend over time: at the beginning of the reference period, most of the drainage basins had medium HV levels and the largest number of high or very high HV, while in 2000–2009 a significant part of the medium HV watersheds turned into low HV units, and in some cases, the very high HV of particular catchments reduced to high, medium, or even low HV. In the latter part of 1990–2018, most of the drainage basins became low HV units, only one catchment turned from a high HV unit to one with very high values of the parameter (the Negrișoara) and only five drainage basins registered high levels of HV.

The southern Bistrița catchment maintained a low HV level through all three periods, while the Negrișoara, Țibău, Cârlibaba, Bistricioara, Northern Bistrița, Bârnaru, Moldovița, and the Northern Moldova maintained HV levels above the medium values for at least two of the three periods. The better part of the drainage basins with high or very high HV were located in the west or northwest of the reference area, corresponding to the tributaries on the right side of the Siret River. On the contrary, the tributaries that joined the Prut from the right side tended to register low or medium HV levels.

The attenuation of the HV through time follows an East–West direction, from the lower elevations of the Moldavian Plateau to the heights of the Carpathians. Hydrological droughts are more likely to occur in the eastern part of the analyzed territory, while the western section displays a certain propensity for floods. These tendencies are explained by the differences between the climate and landform particularities of the two zones.

Knowledge regarding HV levels is vital for protecting human communities in Suceava, Botoșani, Neamț, and Iași counties from the potential destructive effects of floods or hydrological droughts. In the study area, the most important floods occurred in 1991, 1995, 2005, and 2010 [14], and the most consistent hydrological droughts occurred in 2000, 2007, and 2012 [15]. The assessment is useful in the endeavor of elaborating hydrological hazard mitigation strategies, being the first research work of this kind that focuses on the proposed study area. Its results may be used by local and national authorities in order to elaborate more effective strategies that deal with hydrological risks and to develop more targeted water management projects, properly calibrated to the conditions of the Northeast Development Region in Romania. The overview of the considered factors may help detect environmental changes that may increase HV in the future, such as in the case of significant land use changes or of the frequencies of different hydrological hazards (i.e., floods, hydrological droughts).

This paper proves that MCDM methods can be combined and effectively used to assess the HV levels of catchments in a given area, considering factors related to hydrological phenomena frequency, hydrotechnical structure coverage, land use, relief, and hydrological network aspects, of both benefit and non-benefit types. Moreover, the proposed AHP-TOPSIS-based framework may be used to assess different types of environmental vulnerabilities, serving as a practical and adjustable research instrument.

**Author Contributions:** A.-C.A.: conceptualization, methodology, software, writing—original draft preparation. I.M.: data curation, visualization, investigation, supervision. D.B.: data curation, software, validation, investigation. D.L.: writing—reviewing and editing, visualization, supervision. All authors have contributed equally to the present article. All authors have read and agreed to the published version of the manuscript.

**Funding:** This research was financed by the Department of Geography, Faculty of Geography, and Geology, "Alexandru Ioan Cuza" University of Iasi, Romania. Daniel Boicu acknowledges infrastructure support from the Operational Program Competitiveness 2014–2020, Axis 1, under POC/448/1/1 research infrastructure projects for public R&D institutions/Sections F 2018, through the Research Center with Integrated Techniques for Atmospheric Aerosol Investigation in Romania (RECENT AIR) project, under grant agreement MySMIS no. 127324.

**Institutional Review Board Statement:** Not applicable.

**Informed Consent Statement:** Not applicable.

**Data Availability Statement:** The datasets from this study can be accessed upon reasonable request from one of the first two authors.

**Conflicts of Interest:** The authors declare no conflict of interest. The funders had no role in the design of the study; in the collection, analyses, or interpretation of data; in the writing of the manuscript, or in the decision to publish the results.

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
