# Peer review of "Comparative Multi-Criteria Assessment of Hydrological Vulnerability—Case Study: Drainage Basins in the Northeast Region of Romania"

_water, doi:10.3390/w14081302_

Round 1

Reviewer 1 Report

General comments: 

The submitted paper is an interesting approach to the comparison of multi-criteria assessment of hydrological vulnerability, with an application to Romanian territories. The study is correctly conducted, the statistical analysis is well grounded, and the results are significant and with scientific soundness. 

The structure of the manuscript is polished and well written but needing an additional review by a native editing service. However, there are still some lacks, which would need extra amendments.   

Structure of the manuscript 

Abstract 

I found the abstract adequate and concise.  

Introduction 

This section is very well written, and it establishes the main hypothesis, objectives and it even includes the novelty of this study. Congratulations. 

Study area 

Please check units in line 100. This problem is repeated along the whole text. 

Methodology 

I suggest including a well justified basis for the selection of these 2 MCDM methods and no other.  

Please, include the sources of your data and variables. 

The authors must include a justification for the selection of the 3 factors of HV and their criteria. They have no scientific basis.  

It is not clear how you compute some of the criteria for the 3 factors. The authors must include a full description of calculations or how they are produced. 

It is not clear how many experts have assessed the AHP and TOPSIS methods. Usually, there is a round of experts who assess the criteria, and they are grouped based on their knowledge. The authors should include a full description of the experts and the group they belong to. 

In addition, there is no information about the consistency for the experts’ evaluation of the proposed methods.  

Results 

The results are completely conditioned by the number of experts for the assessment of the proposed MCMD methods. The authors must include more information regarding this aspect. 

Conclusions 

This section provides significant conclusions about this work. I suggest enlarging planning implications. 

Author Response

We thank the reviewers for taking the time to read the paper and suggest way to improve it. Here are our answers to their comments. Also, we are grateful to the Editor for managing the peer-review process of our study.

Italics – reviewer’s comment

Normal text – our response

Reviewer 1

General comments: 

The submitted paper is an interesting approach to the comparison of multi-criteria assessment of hydrological vulnerability, with an application to Romanian territories. The study is correctly conducted, the statistical analysis is well grounded, and the results are significant and with scientific soundness. The structure of the manuscript is polished and well written but needing an additional review by a native editing service. However, there are still some lacks, which would need extra amendments.   

Structure of the manuscript 

Abstract 

I found the abstract adequate and concise. Thank you.

Introduction 

This section is very well written, and it establishes the main hypothesis, objectives and it even includes the novelty of this study. Congratulations. Thank you.

Study area 

Please check units in line 100. This problem is repeated along the whole text. 

We corrected this issue. It was a copy-paste problem within the MDPI template.

Methodology 

I suggest including a well justified basis for the selection of these 2 MCDM methods and no other.  

This is specified as follows: “By combining these two MCDM methods, two main aspects of the assessment in question are addressed: the task of establishing the relative importance of the factors at category level is fulfilled by AHP and the necessity to rank a large number of alter-natives is solved by implementing TOPSIS. While other factor weighing MCDM methods could have been used, AHP was preferred because the proposed framework does not take into consideration the relations between the aforementioned factors – situation for which ANP would have been adequate, but uses a structure of main criteria and sub-criteria and integrates and a category level pairwise comparisons approach. Also, TOPSIS is a robust ranking method that can evaluate alternatives with performance scores that are expressed using different measuring units. This combination allows for a salient MCDM assessment of the HV in the area of interest.”

Please, include the sources of your data and variables.

All the data sources are detailed at the beginning of each section that describes the Hydrological, Land use and Constant factors.

The authors must include a justification for the selection of the 3 factors of HV and their criteria. They have no scientific basis.

Each of the integrated factors influence hydrological vulnerability in certain ways, which are explained in the designated sections of the 3 categories of factors. We preferred to argument the contribution/influence of each factor to/on the emergence of the overall hydrological vulnerability, rather than to provide a broad, vague description of their influence. These explanations are accompanied by proper references.

Examples:
- “It should be highlighted that floods and hydrological droughts are the main hydrological phenomena analysed in order to evaluate the HV of a region or a drainage basin. The dynamics of these phenomena [18], together with the one of the climatic parameters [19] are the basis of decisions regarding water resources management [20]. Therefore, the hydrological factors have a relative importance of 50% in the present assessment.”

-“ In several cases, the hydro-technical structures serve multiple purposes, among which regulating the streamflow and attenuating flood waves are the most common. They also reduce the impact of extreme hydrological phenomena like floods and hydrological droughts on human communities. The paradigm implies that this type of structures rarely covers the whole drainage basin [26].”

-These factors relate to the impact of different land use on the hydrological dynamics of the drainage basins. Forests fulfil various ecosystem services, of particular importance in this context being the regulation and provisional ones [27, 28]. The in-fluence of forest areas over flood hazards has been largely studied, and their moderator role on hydrological and climatic processes has been reiterated [16, 29, 30]. Larger forest areas imply that the impact of dangerous hydrological phenomena is delayed and attenuated, as tree canopies and radicular systems regulate streamflow and absorb part of the rainfall. In addition, the effects of settlement areas’ particularities on the hydrological dynamics of the drainage basins are of vital importance [31, 32]. Settlement areas are usually covered by hardened surfaces of concrete, which preclude water infiltration. The same areas may have sewage systems that significantly disrupt infiltration and streamflow patterns.”

-“ The constant factors category includes those indicators that directly and indirectly influence the evolution of hydrological phenomena that impose risks, and that slightly change over time. The slope, drainage density and the circularity coefficient of drain-age basins tend to remain constant, as the processes that lead to their modifications usually require extended periods of time, and only in particular cases they act in forceful and definite manners.”

-“ The geology of the study area is also worth considering, but such factors are not included in the assessment. The geological structure of the Eastern part of Romania generated a series of specific development conditions of the drainage basins. The East-Carpathian Crystalline-Mesozoic Zone [35], with hard rocks of low permeability contributed to the formation of elongated catchments, developed along fault lines. In the platform area, towards the Northeast part of Romania, the Sarmatian and Qua-ternary geological deposits, made of clay, marls and sands with significant thickness, generated extensive drainage basins [36, 37].

Slope (SLOPE) is a static factor that influences HV because it is directly related to the relief. Its values are automatically generated using an ASTER DEM model of 80 m. 36% of the drainage basins are characterised by low slope values (0-5 degrees) that favour slow/diminished streamflow along the sides of the drainage basins and allow for water deposition in the small depressions of the topographic surface or in floodplains.”

-“ Hydrological network density (NETDENS) represents the total length of the river channel that is permanently covered by streamflow divided by the area of the drainage basin. It influences the streamflow volume, as higher values favour the occurrence of floods, with negative impact on the environment [39]. Also, lower values of this factor are correlated with hydrological droughts [40], which are more frequent in the drain-age basins located in the eastern part of the study area.

Circularity ratio (CR) ranks among the morphometric parameters that condition the occurrence and the evolution of hydrological phenomena that constitute threats to the environment and human communities.”

It is not clear how you compute some of the criteria for the 3 factors. The authors must include a full description of calculations or how they are produced.

This is included within the manuscript. For each factor, the data sources and the computation procedures are specified in Methodology.

It is not clear how many experts have assessed the AHP and TOPSIS methods. Usually, there is a round of experts who assess the criteria, and they are grouped based on their knowledge. The authors should include a full description of the experts and the group they belong to.

In addition, there is no information about the consistency for the experts’ evaluation of the proposed methods.  

This requirement was implementing by adding the following clarification in the manuscript: “The weighing of the factors is based on the judgments of a team of four experts knowledgeable on hydrology, climatology, land cover and forest dynamics. They were grouped according to their expertise field, and each of them was asked to perform the pair-wise technique on the set of indicators. Their combined expertise ensures a proper assessment of the contribution of the hydrological, land use and constant factors to the overall hydrological vulnerability of the study area. The judgments were checked for consistency, in order to avoid an erroneous implementation of the MCDM methods. Further on, the final weights were obtained by comparing the results; each expert supporting his/her opinion with on-point arguments. This discussion forum motivated the final weights of the factors, also serving as a consistency-checking instrument.”

Results 

The results are completely conditioned by the number of experts for the assessment of the proposed MCMD methods. The authors must include more information regarding this aspect. 

Already modified as specified in the Methodology section. This was, indeed, an important issue that called for improvements. We are thankful for pointing it out.

Conclusions 

This section provides significant conclusions about this work. I suggest enlarging planning implications.

The section regarding planning implications was improved as requested.

Reviewer 2 Report

Reviewer’s comments to authors:

Title of the manuscript - "Comparative multi-criteria assessment of hydrological vulnerability. Case study: drainage basins in North Eastern part of Romania"

I have reviewed the manuscript entitled “Comparative multi-criteria assessment of hydrological vulnerability. Case study: drainage basins in North Eastern part of Romania”. The article assessed the hydrological vulnerability level of 81 drainage basins in the NE of Romania for three distinctive time periods (1990-1999, 2000-2009, 2010-2018), using a multi-criteria approach. Indeed there are actually few articles devoted to the hydrological vulnerability. Thus, the article presents an interesting topic. But there are multiple issues which must be resolved before accepting this paper. Here are the comments which must be addressed:

Comments:

  1. The article title needs to modify and there should not be dot (.) after vulnerability.
  2. The main finding of your study is not present in the abstract, must include it.
  3. Introduction is inadequate, lack of recent literature, and need improve with recent literature.
  4. The last paragraph of introduction is not important, I recommend for deleting it.
  5. The section on methods is not well presented and should be revised
  6. Figure 2, 6, and 7 need to redraw. It is very poor quality and unclear
  7. The quality of each figure is not high resolution. Try to enhance all figures throughout the manuscript, minimum 300 dpi
  8. Result is section very poor. Based on Figure 8, the overall very high hydrological vulnerability of 81 basins is decreasing from 1990 to 2018 which is not matched with hydrological factors selected in this study (Figure 3, 4, and 5). I have objection here! need to justify
  9. Discussion (section 4) needs to improve. Here, compare your study findings with other relevant studies and also judge the novelty of your study.

Author Response

Response to review

We thank the reviewers for taking the time to read the paper and suggest way to improve it. Here are our answers to their comments. Also, we are grateful to the Editor for managing the peer-review process of our study.

Italics – reviewer’s comment

Normal text – our response

Reviewer 2

I have reviewed the manuscript entitled “Comparative multi-criteria assessment of hydrological vulnerability. Case study: drainage basins in North Eastern part of Romania”. The article assessed the hydrological vulnerability level of 81 drainage basins in the NE of Romania for three distinctive time periods (1990-1999, 2000-2009, 2010-2018), using a multi-criteria approach. Indeed there are actually few articles devoted to the hydrological vulnerability. Thus, the article presents an interesting topic. But there are multiple issues which must be resolved before accepting this paper. Here are the comments which must be addressed:

Comments:

  1. The article title needs to modify and there should not be dot (.) after vulnerability.

We consider that the current title is the most appropriate for this study. The part after the “.” is vital for the understanding the context of the research work, and what is left without it is too general. In order to comply to this requirement, we ask the review to properly motivate it.

  1. The main finding of your study is not present in the abstract, must include it.

The Abstract was modified as requested.

  1. Introduction is inadequate, lack of recent literature, and need improve with recent literature.

Four more recent references were included in the Introduction.

  1. The last paragraph of introduction is not important, I recommend for deleting it.

We complied to this modification.

  1. The section on methods is not well presented and should be revised.

We consider that the Methodology section is properly set up. It begins with a detailed description of the datasets, continuing with the presentation of the workflow and of the implemented MCDM methods. We improved this section as reviewer 1 required, also adding a second introductory paragraph.

  1. Figure 2, 6, and 7 need to redraw. It is very poor quality and unclear.
  2. The quality of each figure is not high resolution. Try to enhance all figures throughout the manuscript, minimum 300 dpi.

We improved the evolution of all the figures and reuploaded them.

  1. Result is section very poor. Based on Figure 8, the overall very high hydrological vulnerability of 81 basins is decreasing from 1990 to 2018 which is not matched with hydrological factors selected in this study (Figure 3, 4, and 5). I have objection here! need to justify.

The reviewer should to understand that AHP and TOPSIS do not work in a linear manner, so to expect that the variation of the considered factors dictates the overall vulnerability evolution is erroneous. The factors do not equally contribute to the emergence of the overall hydrological vulnerability, but are weighed via AHP. These weights are subsequently integrated into the TOPSIS algorithms, which does not follow a linear ranking path. We recommend him/her to reread the Methodology section in order to understand how these methods work and what is the effect of the weighted factors on the overall vulnerability.

  1. Discussion (section 4) needs to improve. Here, compare your study findings with other relevant studies and also judge the novelty of your study.

As presented in this section, we cannot compare the results with other findings, as it is the first study of this type on the study area. Also, the novelty is already explained (lines 511-526, initial pdf):

“This case study on the drainage basins in NE Romania may be considered a trail-blazing one, given the fact that the combination of the two MCDM methods has not been used before. Their suitability is proven by the capacity of such methods to integrate a set of criteria that have different natures and contrasting influences on HV, and also by the fact that they allow the evaluation of a high number of alternatives. Moreover, by applying the framework on three time periods, the case study meets the demands formulated by [5], namely that HV assessments should be performed frequently, due to the changes of the underlying natural and socio-economic conditions specific to drainage basins.

Another element of novelty is that the study area has not been assessed from this point of view before, the present assessment serving as one of the first instruments that may help the elaboration of better hydrological hazard mitigation planning. This means that the results of the assessments cannot be compared to other of similar types, as there are no such assessments targeting the considered study area. Thus, the results may contribute to the development of water management and hydrological risk reduction plans that are adjusted to the particularities of the Northeast Development Region.”

Round 2

Reviewer 1 Report

Thank you for adressing the main issues of the review.

Author Response

We are grateful for taking the time to review our paper. At the same time, we are sorry for making you put additional effort into the work concerning its publication. Indeed, the contradiction of the first two reviewers is what caused a great confusion to us and to the way we modified the article. However, your suggestions are clearer and we hope that we properly complied to your requests.

We rewrote (and extended) the Discussion section, and addressed the importance of the aim from an international point of view. We hope the coherence of the presented ideas was improved. As it would be inappropriate in terms of efficiency to past an entire section in this letter, we kindly ask the Editor to check for modifications in the reviewed manuscript.

This concern was also briefly addressed in the Introduction:

“Nonetheless, the proposed framework aims to contribute to the field of HV analysis by bringing to light the necessity to integrate multiple types of vulnerability (i.e., flood vulnerability, hydrological drought vulnerability); an approach which is often dismissed, to the benefit of focusing on single hazard vulnerability. This is of particular importance when considering the climate changes and the subsequent alteration of hydrological processes and phenomena. The multi-temporal approach is also beneficial to the integration of such interactions in HV assessment, as climatic and hydrological modifications may not be observed when analysed in short time frames. The methodological framework may be replicated focusing on other study areas, but it should be emphasized that the suitability of the indicators may be altered. In order to avoid this, the methodology needs to be applied focusing on catchments that present various elevations, slopes, hydrological network densities, and that are affected by both floods and hydrological droughts, depending on the season and exact location.”

Also, we tried to clarify the reasons why we chose to apply Multi-Criteria Decision-Making (MCDM) methods to assess hydrological vulnerability, and the strong points of the proposed methodological framework.

Reviewer 2 Report

After careful reading the revised manuscript, I found that the authors have not considered  my serious suggestions/observations in their revised manuscript. The authors think they have done a novel work, although the article has some flaws with reference to methodology and results.

Therefore, with regret, I am not able to accept this paper for publication.

Thank you.

Author Response

We are grateful for taking the time to review our paper. At the same time, we are sorry for making you put additional effort into the work concerning its publication. Indeed, the contradiction of the first two reviewers is what caused a great confusion to us and to the way we modified the article. However, your suggestions are clearer and we hope that we properly complied to your requests.

The first issue was addressed by rewriting the Discussion section. In order to provide the background of hydrological vulnerability research, we included 11 more recent papers (published between 2012 and 2021), and modified the idea flow of the Discussion.

We modified the Methodology section and tried to make it “friendlier” to the readers that are not accustomed to MCDM methods or have little knowledge of them. To this end, we added additional comments that should clarify how AHP and TOPSIS work considering the problem at hand (the estimation of the hydrological vulnerability specific to the 81 catchments in NE Romania). We moved the methodological framework diagram at the beginning of this section in order to provide a clear picture of the workflow right from the start. We kindly ask the Editor to review the Methodology section, hoping that we properly addressed the issue, as we had the best intentions to do so.

Sincerely,

The Authors